# Advantages of the Use of Axial Traction Magnetic Resonance Imaging (MRI) of the Shoulder in Patients with Suspected Rota-Tor Cuff Tears: An Exploratory Pilot Study

**DOI:** 10.3390/healthcare11050724

**Published:** 2023-03-01

**Authors:** Roberto Minici, Michele Mercurio, Bruno Iannò, Olimpio Galasso, Giorgio Gasparini, Domenico Laganà

**Affiliations:** 1Radiology Unit, Department of Experimental and Clinical Medicine, “Magna Græcia” University, Mater Domini University Hospital, 88100 Catanzaro, Italy; 2Department of Orthopaedic and Trauma Surgery, “Magna Græcia” University, Mater Domini University Hospital, 88100 Catanzaro, Italy; 3Department of Surgery, Division of Orthopedics and Trauma Surgery, “G. Jazzolino” Hospital, Piazza Fleming, 89900 Vibo Valentia, Italy

**Keywords:** axial traction MRI, axial load MRI, MRI, shoulder mri, rotator cuff tears

## Abstract

Magnetic Resonance Imaging (MRI) with axial traction is a tool for the assessment of musculoskeletal pathology. Previous reports have demonstrated a better distribution of intra-articular contrast material. No investigations were performed to evaluate glenohumeral joint axial traction MRI in patients with suspected rotator cuff tears. This study aims to assess the morphological changes and the potential advantage of glenohumeral joint axial traction MRI without intra-articular contrast administration in patients with suspected rotator cuff tears. Eleven patients with clinical suspicion of rotator cuff tears underwent a shoulder MRI scan with and without axial traction. PD weighted images with SPAIR fat saturation technique and T1 weighted images with TSE technique were acquired in the oblique coronal, oblique sagittal and axial planes. Axial traction allowed a significant widening of the subacromial space (11.1 ± 1.5 mm vs. 11.3 ± 1.8 mm; *p* = 0.001) and inferior glenohumeral space (8.6 ± 3.8 mm vs. 8.9 ± 2.8 mm; *p* = 0.029). With axial traction, there was a significant decrease in measurements of the acromial angle (8.3 ± 10.8° vs. 6.4 ± 9.8°; *p* < 0.001) and gleno-acromial angle (81 ± 12.8° vs. 80.7 ± 11.5°; *p* = 0.020). Our investigation demonstrates for the first time significant morphological changes in the shoulder of patients with suspected rotator cuff tears who underwent a glenohumeral joint axial traction MRI.

## 1. Introduction

Magnetic Resonance Imaging (MRI) with axial traction or dynamic positioning is a tool for the assessment of musculoskeletal pathology [1,2,3,4,5]. Diagnostic accuracy may be improved due to the morphological changes in the investigated joint [2,6,7]. Axial traction MRI of the wrist, elbow, hip, knee and ankle are commonly associated with arthrography, allowing a widening of the joint spaces, a better distribution of intra-articular contrast material and improved visualization of intra-articular structures [7,8,9,10].

The application of axial traction to MRI of the shoulder is a limited but active area of research [1,10,11,12]. Chan et al. found an improved detection rate of superior labrum anterior and posterior (SLAP) tears with a combination of MR arthrography and axial traction in a cadaveric series [11]. Becce et al. found that MR arthrography of the shoulder under axial traction increases subacromial and glenohumeral joint space widths, and prompts better coverage of the superior labrum-biceps tendon complex and articular cartilage by contrast material [10]. Jung et al. demonstrated an improved specificity by a better distinction of true pathology from labral anatomic variants [13]. In these studies, a combination of MR arthrography and axial shoulder traction was used. More recently, Garwood et al. evaluated the effect of axial traction on the glenohumeral joint without intra-articular contrast administration [1], demonstrating that glenohumeral joint axial traction MRI is technically feasible and well tolerated; traction of the capsule, widening of the superior glenohumeral joint space and acromial angle were observed. This investigation was performed with a 3T MRI unit on healthy volunteers.

MRI has shown high diagnostic accuracy for full-thickness tears of the rotator cuff, but suboptimal performance remains for partial thickness tears. Brockmeyer et al. found that the MRI sensitivity in identifying partial-thickness tears was 51.6%, specificity 77.2%, positive predictive value 41.3% and negative predictive value 83.7% [14]. New techniques applied to MRI seem necessary to improve accuracy and no investigations have been performed to evaluate glenohumeral joint axial traction MRI in patients with suspected rotator cuff tears. This study aimed to assess the morphological changes of the shoulder by performing a glenohumeral joint axial traction MRI without intra-articular contrast administration in patients with suspected rotator cuff tears.

## 2. Materials and Methods

This study is a single-center analysis of prospectively collected data from consecutive patients who had undergone, from January 2020 to May 2022, a shoulder MRI scan with and without axial traction for suspected rotator cuff tear. The study protocol was approved by the local ethics committee, and the research was conducted in compliance with the Declaration of Helsinki (ID 05/2015). Inclusion criteria were (1) clinical suspicion of rotator cuff tears; (2) execution of a shoulder MRI scan, with and without axial traction; (3) age between 18 and 65 years; (4) no previous shoulder surgery; (5) willingness to participate in the study. The exclusion criteria were (1) neurological disorders of the upper extremities; (2) concomitant rheumatological diseases; (3) previous rotator cuff tears on the same shoulder. Informed consent was obtained from all individual participants included in the study after the nature of the procedure had been fully explained.

All patients underwent clinical evaluation by trained physicians who were not involved in the primary care of the patients. Constant-Murley score (CMS), flexion-extension and abduction-adduction range of motion (ROM) were assessed. The Jobe, external rotation lag sign (ERLS), Lift-off, Hawkins-Kennedy, O’Brien and Palm-up tests were performed.

MRI scans were performed using a 1.5 T unit (Achieva XR, Philips Medical System, Eindhoven, The Netherland) with a dedicated eight-channel shoulder coil (Philips SENSE Shoulder coil 11-elements). Patients were in a supine position with the arm adducted to 15° of external rotation. Coronal oblique images were oriented parallel to the supraspinatus muscle. Sagittal oblique images were oriented perpendicular to the supraspinatus muscle. A section thickness of 3 mm was used for all images. A three-plane localizer was taken at the beginning of the exam. The examination protocol included two consecutive phases. The first involved a shoulder scan without axial load. Proton density-weighted (PDW) images with SPAIR fat saturation technique were acquired in the oblique coronal, oblique sagittal and axial planes. T1 weighted (T1W) images with turbo spin echo (TSE) technique were acquired in the oblique coronal and sagittal coronal planes. The second phase was performed in the same session and consisted of an identical acquisition protocol, with the addition of MR-compatible axial traction, avoiding patient’s repositioning. Non-adhesive skin traction was used. First, the traction kit was folded around the wrist and hand, with the foam innermost and the stirrup plate over the clenched fist. It was then fixed by applying an elastic bandage starting at the wrist and continuing up the arm using the ascending spiral technique. Two straps depart from the skin traction device parallel to the long axis of the body and slide on a pulley, attaching to a weight of 4 kg. The patients were instructed to relax and not actively resist the applied traction force, as in Garwood et al. [1] and Becce et al. [10]. The total duration of time each subject spent with axial traction applied throughout positioning in the MRI gantry and image acquisition was approximately 10 min. In case of motion artifacts caused by breathing or involuntary movements of the patients, the sequence was re-acquired to avoid diagnostic impairment.

Two radiologists (R.M. and D.L.), with 4 and 30 years of experience in musculoskeletal radiology respectively, evaluated every MR examination (BrainLAB, Munich, Germany), blinded to axial traction application. Subacromial space, superior glenohumeral joint space, inferior glenohumeral joint space, acromial angle and gleno-acromial angle were calculated (Figure 1). Distance between opposing cortical surfaces defines the joint spaces. The inferior glenohumeral joint space was measured at a level 10 mm above the inferior glenoid rim and the superior glenohumeral joint space was measured at a level 5 mm below the superior glenoid rim. Coronal images were used to calculate glenohumeral joint spaces [10]. The acromial angle was defined by the angle between the inferior surface of the acromion and the horizontal line, as in Garwood et al. [1]. The gleno-acromial angle was defined by Banas et al. [15] as the lateral acromial angle and was identified as the angle between the inferior surface of the proximal end of the acromion and the glenoid face on the oblique coronal plane images, just posterior to the acromioclavicular joint. The subacromial space was evaluated on T1-weighted coronal oblique images, excluding the thicknesses of the articular cartilage from the measurements, thus taking into account the true gliding space left for the supraspinatus tendon and avoiding limitations related to X ray-based measurements [16].

All data were measured, collected, and reported with one decimal point accuracy. Data were maintained in an Excel spreadsheet (Microsoft Inc., Redmond, WA, USA) and the statistical analyses were performed using SPSS software (SPSS, version 22 for Windows; SPSS Inc, Chicago, IL, USA) and R/R Studio (Boston, MA, USA) software. The analysis of efficacy was based on a per-protocol basis. Kolmogorov-Smirnov test and Shapiro-Wilk test were used to verify the normality assumption of data. The mean and standard deviation were noted for the continuous variables, as well as counts for the categorical variables. Continuous not normally distributed data are presented as median (interquartile range: 25th and 75th percentiles—IQR). The unpaired Student t-test was used to assess statistical differences for continuous normally distributed data, while categorical and continuous not normally distributed data were assessed using the Chi-squared test and the Mann-Whitney test, respectively. The Wilcoxon signed-rank test was performed to compare matched samples. The mean values of the measurements obtained by the two observers were used to compare standard and axial-traction data. An intraclass correlation coefficient (ICC) was used to determine if items could be rated reliably by different raters. A two-way random effects model was performed, using the absolute agreement as the relationship among raters and the single rater as the unit of interest. Interpretation thresholds for ICC measures followed the guideline given by Koo and Li [17]: poor (<0.50), moderate (0.50–0.75), good (0.75–0.90), excellent (>0.90). A *p*-value of < 0.05 was considered significant.

## 3. Results

The characteristics of the study population are summarized in Table 1. Fourteen patients were initially recruited. Two patients experienced claustrophobia during MRI and asked to stop the exam, whereas one patient with severe obesity did not enter the MRI scanner. Therefore, 11 patients were enrolled and evaluated. Clinical data are detailed in Table 2.

Inter-rater agreement was moderate for measurements of the gleno-acromial angle with traction (ICC 0.744) and inferior glenohumeral space with traction (ICC 0.685). An excellent inter-rater agreement was observed for measurements of the type of tear (ICC 1), Goutallier classification (as proposed by Fuchs et al. [18]) (ICC 1), Patte classification [19] (ICC 1), Warner classification [20] (ICC 1), subacromial space (ICC 0.995), subacromial space with traction (ICC 0.984), superior glenohumeral space (ICC 0.955), superior glenohumeral space with traction (ICC 0.953), acromial angle (ICC 0.995), acromial angle with traction (ICC 0.998) and gleno-acromial angle (ICC 0.795). There was good inter-rater agreement for measurements of the inferior glenohumeral space (ICC 0.848) and gleno-acromial angle (ICC 0.795). All grade 4 injuries according to Goutallier classification were recorded in patients over 50 years of age. Details are given in Table 3.

Axial traction allowed a significant widening of the subacromial space (11.1 ± 1.5 mm vs. 11.3 ± 1.8 mm; *p* = 0.001) and inferior glenohumeral space (8.6 ± 3.8 mm vs. 8.9 ± 2.8 mm; *p* = 0.029). With axial traction, there was a significant decrease in measurements of the acromial angle (8.3 ± 10.8° vs. 6.4 ± 9.8°; *p* < 0.001) and gleno-acromial angle (81 ± 12.8° vs. 80.7 ± 11.5°; *p* = 0.020). No statistically significant difference was observed in measurements of the superior glenohumeral space with and without axial traction (11.6 ± 4.6 mm vs. 9.2 ± 4.9 mm; *p* = 0.074). Outcome data are reported in Table 4.

## 4. Discussion

To the best of our knowledge, this is the first study aimed at assessing the morphological changes in the shoulder of symptomatic patients with suspected rotator cuff tears, who underwent a glenohumeral joint axial traction MRI with a 1.5T MR unit without intra-articular contrast administration. A statistically significant increase in subacromial space and inferior glenohumeral space was observed with axial traction, with a good agreement between raters.

The widening of the glenohumeral joint space is associated with multiple advantages. Firstly, chondral surfaces are better depicted with a resolution of their overlap that is usually observed in standard shoulder MRI scans [1]. Hence, axial traction allows us to evaluate the opposing articular surfaces as a distinct entity. In contrast, evaluation of chondral pathologies, such as signal heterogeneity or focal chondral defect, is often limited in neutral positioning by the summation effect regarding the chondral surfaces. Secondly, the amount of the intra-articular joint fluid increases in the glenohumeral space, mimicking the presence of intra-articular contrast, with improved visualization of the glenoid labrum. Shoulder MRI scans without intra-articular contrast administration can hardly distinguish labral pathology and anatomic variants of the glenoid labrum. In cadaveric shoulders, Chan et al. demonstrated an improved detection rate and characterization of SLAP tears with a combination of MR arthrography and axial traction than MR arthrography without axial traction [11]. Becce et al. found that MR arthrography of the shoulder under axial traction prompts better coverage of the superior labrum-biceps tendon complex by contrast material [10]. Other investigations showed an improved identification of labral pathology when an MRI scan is performed with the arm in a stress position than in a neutral [3,21,22]. An improved specificity via a better distinction of true pathology from labral anatomic variants was highlighted by Jung et al. with a combination of MR arthrography and external rotation [13]. More recently, Garwood et al. have evaluated the effect of axial traction on the glenohumeral joint in the absence of intra-articular contrast administration with a 3T MR unit in healthy volunteers. The authors demonstrated a widening of the glenohumeral joint space with three incidental previously undiagnosed superior labral tears, thus hypothesizing an improved characterization of labral pathology versus anatomic variants [1]. Despite this hypothesis going beyond the aims of our investigation, according to these data, the axial load may improve diagnostic ease and accuracy of labral tears or, at least, can lead to suspicion of labral pathology with greater confidence, better choosing which patients with shoulder pain to refer for MRI arthrography. Finally, we hypothesize that glenohumeral joint axial traction MRI without intra-articular contrast administration may improve the evaluation of chondral and labral pathology by widening the glenohumeral joint space.

The widening of the subacromial space allows better visualization of the subacromial structures, in particular the supraspinatus tendon. Other investigations performed in vivo have shown similar results. Becce et al. found a statistically significant increase in the subacromial space in a MR arthrography series of the shoulder under axial traction [10]. Garwood et al. confirmed this finding in healthy volunteers without reaching the statistical significance, in the absence of intra-articular contrast administration and with a 3T magnetic resonance [1]. The hypothesis that a widening of the subacromial space is associated with a better detection rate of partial thickness tears of the supraspinatus tendon has not been tested by our study. It would be useful to evaluate this hypothesis in future investigations, considering that the diagnosis of partial thickness tears of the rotator cuff by clinical tests and MRI procedures remains challenging. Brockmeyer et al. have recently highlighted that the diagnostic accuracy of MRI, clinical tests (Jobe-test and Impingement-sign) alone and a combination of both is limited for detecting partial thickness tears of the rotator cuff. For the combination of MR imaging, Jobe-test and Impingement-sign, sensitivity was only 46.9% [14]. Recent studies have focused on the utility of radial MRI of the shoulder to significantly improve the assessment of rotator cuff tears [23,24]. Furthermore, Banas et al. demonstrated a statistically significant correlation between the gleno-acromial angle (also called lateral acromion angle) and MRI-detected rotator cuff disease; in particular, as the gleno-acromial angle decreased, a statistically significant increase in rotator cuff disease was noted (*p* < 0.001) [15]. In our study, a statistically significant decrease in gleno-acromial angle is recorded when axial traction is applied to the shoulder MRI scan. Another speculation is that a better visualization of the supraspinatus tendon may allow for better surgical planning. Lippe et al. showed that the Goutallier, Patte, and Warner MRI classification systems for describing rotator cuff tears did not have high interobserver reliability among three experienced orthopedic surgeons [25]. In addition, 3D dual echo-time T1-weighted FLASH sequence has been developed to perform a quantitative MRI assessment of glenoid bone loss, as an alternative to traditional CT reconstructions [26,27]. More recently, the use of 3D MRI reconstructions of the rotator cuff showed significant improvement in accuracy in determining the shape of rotator cuff tears, compared with traditional 2D imaging [28,29]. The surgical treatment of symptomatic partial thickness tears of the supraspinatus tendon is controversial, with different arthroscopic treatment options depending on the extent of the tear, quality of the tendon and individual patient characteristics [30], so that the aforementioned preoperative imaging classifications can play a pivotal role in better surgical planning. This speculation goes beyond the scope of our study but is worth noting.

A good or excellent inter-rater agreement was observed for all quantitative measurements, except for moderate concordance for measurement of the gleno-acromial angle with traction and inferior glenohumeral space with traction. Based on these data, it can be assumed that there is a good consensus in the ratings given by observers. Both radiologists were experienced in musculoskeletal radiology, but with a marked gap in years of experience. Despite this difference, shoulder evaluation with axial traction MRI is highly reproducible. Therefore, glenohumeral joint axial traction MRI with a 1.5T MR unit without intra-articular contrast administration produces data that are reliable for clinical and research purposes. Furthermore, glenohumeral joint axial traction MRI is technically feasible and well tolerated, according to Garwood et al. [1].

The major limitations of the current study are the lack of a control group, the single-center setting, the small study population, and the limited literature, necessary to evaluate the congruence and the consistency of the data presented. However, the present study was performed as an exploratory pilot study and it should be considered as a generator of hypotheses, which should be tested in future investigations to better understand and fully explore the diagnostic potential of axial traction of the glenohumeral joint without intra-articular contrast administration on the 1.5 T MRI unit for suspected rotator cuff tear.

## 5. Conclusions

Our investigation demonstrates for the first time significant morphological changes in the shoulder of patients with suspected rotator cuff tears who underwent a glenohumeral joint axial traction MRI without intra-articular contrast administration. Statistically significant increases in inferior glenohumeral space and subacromial space and a decrease in gleno-acromial angle were noted. Further investigations are needed to test if the axial traction MRI of the shoulder joint without intra-articular contrast administration is associated with an improved detection rate of chondral, labral and rotator cuff pathologies.

## Figures and Tables

**Figure 1 healthcare-11-00724-f001:**
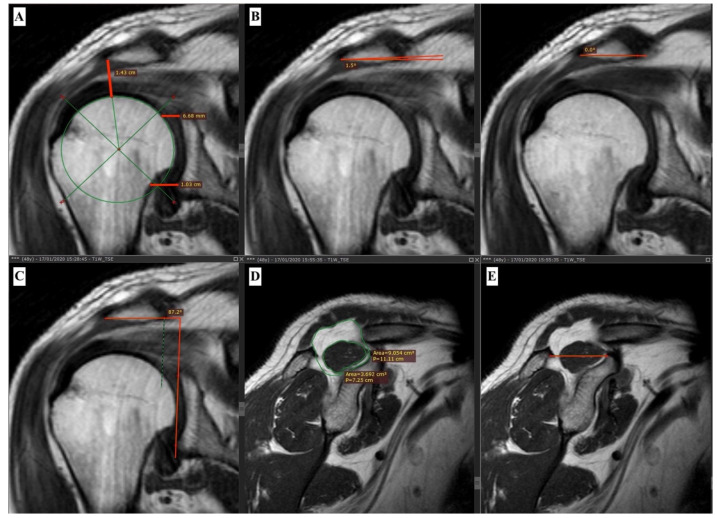
A 48-year-old man with clinical suspicion of rotator cuff injury. T1 weighted (T1W) images with turbo spin echo (TSE) technique were acquired in the oblique coronal and sagittal coronal planes. Measurements of the subacromial space, superior and inferior glenohumeral joint spaces (**A**), acromial angle with and without axial traction (**B**), gleno-acromial angle (**C**), Goutallier classification (**D**) and Warner classification (**E**).

**Table 1 healthcare-11-00724-t001:** Population data.

Variables	
Age (years)	52.4 (±11.6)
Sex (M/F)	5 (45.4%)/6 (54.6%)
Height	164 (±11.4)
Weight	75.8 (±13.5)

**Table 2 healthcare-11-00724-t002:** Clinical data.

Variables	
Traumatic injury (yes/no)	5 (45.4%)/6 (54.6%)
Dominant arm (yes/no)	9 (81.8%)/2 (18.2%)
Flexion-extension ROM	150° (70°)
Abduction-adduction ROM	120° (70°)
Jobe Test (positive/negative)	9 (81.8%)/2 (18.2%)
ERLS Test (positive/negative)	2 (18.2%)/9 (81.8%)
Lift-off Test (positive/negative)	6 (54.6%)/5 (45.4%)
Hawkins-Kennedy Test (positive/negative)	5 (45.4%)/6 (54.6%)
O’Brien Test (positive/negative)	4 (36.3%)/7 (63.7%)
Palm-up Test (positive/negative)	9 (81.8%)/2 (18.2%)
Constant Murley Score	57 (±22.2)

**Table 3 healthcare-11-00724-t003:** Imaging data and intraclass correlation with and without axial traction.

Variables	First Rater	Second Rater	Intraclass Correlation Coefficient (ICC)
Rotator cuff tear (full/partial)	36.4%/63.6%	36.4%/63.6%	1
Goutallier (4/3/2/1/0)	54.6%/18.2%/18.2%/9%/0	54.6%/18.2%/18.2%/9%/0	1
Patte (1/2/3)	100%/0/0	100%/0/0	1
Warner (0/1/2/3)	54.6%/36.4%/9%/0	54.6%/36.4%/9%/0	1
Subacromial space	11.2 (4.2)	11.1 (1.4)	0.995
Subacromial space (traction)	12 (3.5)	11.2 (1.2)	0.984
Superior Glenohumeral space	11.7 (4.7)	11.5 (4.2)	0.955
Superior Glenohumeral space (traction)	9.2 (3.6)	9.2 (5.9)	0.953
Inferior Glenohumeral space	8.1 (3.8)	8.7 (4.2)	0.848
Inferior Glenohumeral space (traction)	8.8 (2)	8.9 (3.3)	0.685
Acromial angle	8.4 (10.2)	8 (10.4)	0.995
Acromial angle (traction)	6.8 (9.6)	5.9 (9.3)	0.998
Glenoacromial angle	80.7 (12)	81.3 (11.5)	0.795
Glenoacromial angle (traction)	80.2 (10.3)	82.9 (10.8)	0.744

**Table 4 healthcare-11-00724-t004:** Outcome data.

Variables	Standard MRI	MRI with Axial Traction	*p* Value
Subacromial space	11.1 (±1.5)	11.3 (±1.8)	0.001
Superior Glenohumeral space	11.6 (±4.6)	9.2 (±4.9)	0.074
Inferior Glenohumeral space	8.6 (±3.8)	8.9 (±2.8)	0.029
Acromial angle	8.3 (±10.8)	6.4 (±9.8)	<0.001
Glenoacromial angle	81 (±12.8)	80.7 (±11.5)	0.020

## Data Availability

The data presented in this study are available on request from the corresponding author. The data are not publicly available due to privacy restrictions.

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
