# Peer review of "Advantages of the Use of Axial Traction Magnetic Resonance Imaging (MRI) of the Shoulder in Patients with Suspected Rota-Tor Cuff Tears: An Exploratory Pilot Study"

_healthcare, 2023, doi:10.3390/healthcare11050724_

Round 1

Reviewer 1 Report

In this study, Minici et al assessed suspected rotator cuff tears using glenohumeral joint axial traction MRI without contrast agent. Although the results demonstrated axial traction allowed widening of subacromial space and inferior glenohumeral space accompanied by decrease in acromial angle and glenoacromial angle, this study, as the authors pointed out themselves in discussion, was conducted with significant limitations such as small sample size and lack of control. Nonetheless, this study is a, as the authors suggested, a pilot study and future studies will be needed to assess whether axial traction can indeed improve sensitivity of diagnosis of rotator cuff tears. My detailed comments are follows:

1.       Have the authors account for motion artifact in their study?

2.       Typo in line 186-187.

Author Response

Dear Reviewer,

Thanks for your comments and suggestions. We welcome them fully and we revised the manuscript accordingly.

  1. Have the authors account for motion artifact in their study?

A: The proper description has been added in Materials section, thus highlighting the accounting of motion artifacts.

  1. Typo in line 186-187.

A: We eliminated the error in the revise text.

Reviewer 2 Report

Dear Author

 The article “Axial traction magnetic resonance imaging (MRI) of the shoul- der in patients with suspected rotator cuff tears: an exploratory  pilot study”,  is an important article for the development of other investigations in the area.

The summary presents all the expected elements as well as the introduction. The material and methods chapter explains in a very objective way all the procedures and criteria associated with the study. In the results, table 1 (sex (MF) does not show the 11 patients (4+5) nor the percentage of 4 (44.4%) and 5 (55.6%) are correct. It should be corrected.

It is not understood why in the IMC is shown in table 1, since it has no associated data, nor is it referred to in the method that IMC was calculated. The discussion is not very robust, due to the lack of bibliography. But the limitations of the study and conclusions are well defined.

Considering all the content of the article, I suggest that the title be changed to “Advantages of the use of Axial traction magnetic resonance imaging (MRI) of the shoul-2 der in patients with suspected rotator cuff tears: an exploratory pilot study”

Author Response

Dear Reviewer,

Thanks for your comments and suggestions. We welcome them fully and we revised the manuscript accordingly.

  1. The summary presents all the expected elements as well as the introduction. The material and methods chapter explains in a very objective way all the procedures and criteria associated with the study. In the results, table 1 (sex (MF) does not show the 11 patients (4+5) nor the percentage of 4 (44.4%) and 5 (55.6%) are correct. It should be corrected.

A: Typos you’ve noted have been corrected.

  1. It is not understood why in the IMC is shown in table 1, since it has no associated data, nor is it referred to in the method that IMC was calculated. The discussion is not very robust, due to the lack of bibliography. But the limitations of the study and conclusions are well defined.

A: IMC has been removed in the revised text according to your comment.

  1. Considering all the content of the article, I suggest that the title be changed to “Advantages of the use of Axial traction magnetic resonance imaging (MRI) of the shoul-2 der in patients with suspected rotator cuff tears: an exploratory pilot study”

A: Manuscript’s title has been updated.

Reviewer 3 Report

The article is well written and clear. References are appropriate, although some should be more recent.

Here are some thoughts.

The number of patients outside the abstract should be clearly stated in the article.

You report that MRI has average sensitivity and specificity, and axial traction MRI of the glenohumeral joint without intra-articular contrast has better results. Complete your bibliography.

MRI with 3D imaging why is it not mentioned at all?Does it perform better or worse than the proposed technique?

The 11 people in the study are between 18 and 65 years old. Due to the fact that the sample is small and the age difference between the patients is large, consider the parameter that there are differences in imaging between injuries in young subjects and fatty degeneration in older subjects.

Ηow many degrees do you set slight external rotation?

Enter the study approval number.

Author Response

Dear Reviewer,

Thanks for your comments and suggestions. We welcome them fully and we revised the manuscript accordingly.

  1. The number of patients outside the abstract should be clearly stated in the article.

A: The number of patients is now stated in the first lines of Results section of the Abstract.

  1. You report that MRI has average sensitivity and specificity, and axial traction MRI of the glenohumeral joint without intra-articular contrast has better results. Complete your bibliography. MRI with 3D imaging why is it not mentioned at all?Does it perform better or worse than the proposed technique?

A: We added a paragraph about 3D MRI and Radial MRI and completed the bibliography in the revised text.

  1. The 11 people in the study are between 18 and 65 years old. Due to the fact that the sample is small and the age difference between the patients is large, consider the parameter that there are differences in imaging between injuries in young subjects and fatty degeneration in older subjects.

A: Sample size is too small to perform subgroup analysis; according to your comment, we added a sentence in the results section to emphasize differences in imaging features between young and older patients.

  1. Ηow many degrees do you set slight external rotation?

A: We clarified degrees of external rotation in the revised text (fifteen degrees).

  1. Enter the study approval number.

A: We entered the study approval number in the revised text.

Round 2

Reviewer 1 Report

The authors have addressed my comments appropriately.